# Brain–Language Model Alignment: Insights into the Platonic Hypothesis and Intermediate-Layer Advantage

Angela Lopez-Cardona[1,2]   Sebastian Idesis[1]   Mireia Masias-Bruns[1]   Sergi Abadal[2]

Ioannis Arapakis[1]

[1] Telefónica Research, Barcelona, Spain

[2] Universitat Politècnica de Catalunya, Barcelona, Spain

## Abstract

Do brains and language models converge toward the same internal representations of the world? Recent years have seen a rise in studies of neural activations and model alignment. In this work, we review 25 fMRI-based studies published between 2023 and 2025 and explicitly confront their findings with two key hypotheses: (i) the Platonic Representation Hypothesis—that as models scale and improve, they converge to a representation of the real world, and (ii) the Intermediate-Layer Advantage—that intermediate (mid-depth) layers often encode richer, more generalizable features. Our findings provide converging evidence that models and brains may share abstract representational structures, supporting both hypotheses and motivating further research on brain–model alignment.

## 1   Introduction

An emerging field at the intersection of neuroscience and Artificial Intelligence (AI) seeks to address the question: do brains and models converge toward the same internal representation of the world? It studies whether biological brains and artificial models create similar representations when exposed to the same stimuli [1], while the AI field offers new explanations for why such convergence might occur. **Language Models (LMs)** have become central to this hypothesis space, as their emerging capabilities raise questions about whether their internal representations parallel human language processing [2]. Numerous studies have shown structural similarities between brain activations and those of LMs [1]. Investigating these similarities requires a shared representational domain, typically established through partial linear mappings between features extracted from neural recordings—often via functional magnetic resonance imaging (fMRI)—and activations from computational models exposed to the same stimulus [1, 3, 4].

These findings naturally raise a deeper question: what drives this alignment? [3]. Current studies explore, for instance, how alignment varies with LMs performance [5], the relative contribution of linguistic features and perceptual features [6], and the impact of model scale, architecture, or dataset size [7]. A more recent line of research examines whether task-specific fine-tuning, multimodal integration, or human alignment data can systematically enhance brain–model correspondence [8, 9].

One theoretical perspective relevant to this question is the **Platonic Representation Hypothesis (PRH)**, introduced by Huh et al. [10]. It suggests that as neural networks scale and improve, their internal representations converge toward a shared statistical model of reality. This convergence may extend beyond artificial systems: biological systems, such as the human brain, might also share aspects of this abstraction, as both face the same fundamental challenge of efficiently extracting and understanding the underlying structure in images, text, sounds, and other modalities. Building on this idea, we seek evidence in the most recent related works on representation alignment that, if models

Preprint.

are converging toward a representation of the real world, and the brain also represents that same world, then both should converge toward each other.

In theoretical AI, increasing interest in Large Language Models (LLMs) has revealed that the final layers are not always the most informative. For example, Skean et al. [11] presented a comprehensive, layer-wise analysis across architectures, model sizes, and tasks. Their findings suggest that intermediate layers consistently outperform final layers, especially in autoregressive models. Interestingly, intermediate layers in LMs have been found to exhibit greater similarity to brain representations [12].

Bringing together these perspectives, and considering the rapid evolution of the field, we review **25 studies**, all published since 2023, that investigate similarity between brain representations and those of LMs, summarized in Section 3, and comparing our work to other reviews in Section 2. Our goal is to assess whether the evidence supports qualitatively two key hypotheses proposed in prior work: (i) the Platonic Representation Hypothesis (Section 4), and (ii) the Intermediate-Layer Advantage (Section 5). Finally, in Section 6, we summarise the main insights and discuss open questions and limitations.

## 2    Related work

Karamolegkou et al. [3] reviewed over 30 studies published until 2023, comparing evidence on the similarity between LM representations and brain activity. Their analysis suggests that, while the evidence remains inconclusive, correlations with model size and quality offer cautious optimism. More recently, Oota et al. [1] provided a comprehensive survey of brain-model alignment across modalities and tasks, covering encoding/decoding pipelines, datasets, and evaluation choices in depth.

In comparison, our review is intentionally more focused: we exclusively review the most recent fMRI-based studies (2023-2025), and provide more up-to-date insights. We restrict our analysis to fMRI as it is the most widely used modality in this field, enabling easier comparison across studies, and offering higher spatial resolution. From this filtering process, we derived a list of 25 works, which form the basis of our review. Rather than treating alignment as an empirical curiosity, we explicitly use it to test two theoretical emerging AI hypotheses: (i) the Platonic Representation Hypothesis, and (ii) the Intermediate-Layer Advantage. This framing shifts the focus from *"what has been observed?"* to *"do the data support these specific hypotheses?"*

## 3    Summary of Reviewed Works: Data, Models, and Methods

In this section, we provide an overview of the reviewed works, focusing on three key aspects: the research questions driving each study (Subsection 3.1), the datasets and models employed (Subsection 3.2), and the experimental methods used to assess similarity between brain activity and model representations (Subsection 3.3). This synthesis highlights common patterns and methodological variations across the literature. In Section 4 and Section 5, we build upon this overview to present the main conclusions and supporting evidence, mapping each work to the specific hypothesis it addresses.

### 3.1    Main Research Directions

Several studies have investigated brain-LM alignment to address different research questions. We categorize these works according to the specific questions they target, as indicated in Table 1.

### 3.2    Datasets and models

**Datasets**. In Table 2, we detail the datasets used and the models evaluated in each study. The choice of dataset and model varies depending on the research question. Datasets may capture brain activity during natural language processing, visual perception, auditory experiences, or a combination of the above, labeled as **R**(eading), **V**(iewing), and **L**(istening) respectively (see Table 2). Some examples interpolate these datasets with **Eye-tracking (ET)** data to trace the word-to-word transitions during reading, then sum the corresponding fMRI signal over each transition, as in St-Laurent et al. [33].

**Models**. Several studies rely on **text-based** Transformer [34] architectures. These include (i) encoder-only models like BERT [35], which are bidirectional; (ii) decoder-only models, such as GPT [36] and LLaMA [37, 38], and (iii) encoder–decoder models, such as BART [39] or T5 [40]. Other studies use

Table 1: Thematic categorization of reviewed works.

| Theme | Representative question | Works |
|---|---|---|
| Information content in representations | Which linguistic/stimulus features (lexical, syntactic, semantic, stimulus-driven) drive brain–model alignment? | [5, 6, 13–16] |
| Scaling laws and architecture size | How do parameter count, data scale, and architectural choices affect alignment? | [7, 17–20] |
| Task-specific training effects | Do models trained for specific objectives (e.g., moral reasoning, speech) align better with brain data? | [21–24] |
| Instruction-tuning and human alignment | Does instruction tuning change the correspondence between model representations and neural activity? | [8, 17, 19, 25] |
| Cross-lingual and multilingual effects | Do different languages converge to a shared conceptual space in the brain? | [26] |
| Brain-informed tuning | Does fine-tuning on brain/behavioral signals improve neural predictivity? | [21–23, 27] |
| Modality differences | How do audio-based vs. text-based models compare in predicting brain signals? | [7, 28] |
| Multimodal vs. unimodal models | Do multimodal models predict brain activity better than unimodal ones? | [8, 9, 13, 29–32] |

**audio-based** models, including Wav2Vec 2.0 [41] and HuBERT [42], which learn unimodal audio representations. For **vision**-only models, variants of ViT [43] are common, often compared against multimodal models such as CLIP [44], trained to align images and text.

**Multimodal Large Language Models (MLLMs)**, which are LLMs that extend their functionality beyond textual data by training on heterogeneous datasets [4], are also increasingly used. Specially Vision Large Language Models (VLLMs) such as InstructBLIP [45] and mPLUG-Owl [46], which generally follow a LLM architecture augmented with an additional visual encoder. In both LLMs and MLLMs, versions are used both before and after undergoing **human alignment** techniques. These post-training methods aim to align model outputs with human expectations [4]. Both aligned and non-aligned variants are used to study how this post-processing affects correlation with brain representations. The category for each specific model is in Appendix A.1 ( Table 4).

## 3.3 Methods for Brain–Model Alignment

Approaches to assessing brain–model similarity vary across studies. The majority of recent studies (22 of 25) rely on the **encoding model** [1], in contrast to earlier work that examined alternative approaches (see Karamolegkou et al. [3]). As shown in Figure 1, the encoding framework predicts brain activity (e.g., fMRI responses) directly from model representations: the same stimuli are presented to both systems, a linear mapping is trained, and its accuracy (typically measured by correlation) defines the **brain score** [7, 47]. This method underlies many recent studies [8, 18–20, 25, 32].

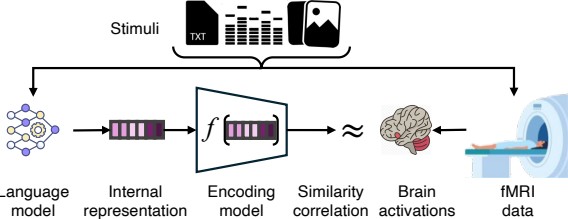

Figure 1: Encoding model framework for brain–model alignment. Model activations are linearly mapped to fMRI responses, and alignment is quantified by correlation.

One methodological variation accounts for the **temporal** nature of the neural signal, modeling the delay of the hemodynamic response. In the temporal approach, model representations are temporally aligned with brain recordings using methods such as Lanczos interpolation [12] together with a

finite impulse response (FIR) model, with multiple lags (e.g., 2–8 s). This is used in works such as [5–7, 16, 17, 21, 23, 24, 26, 27, 29–31]. The **residual approach**, proposed by Toneva et al. [48], predicts brain activity from a baseline model, derives residuals, and evaluates whether an alternative model explains variance in them, thereby isolating unique predictive power. This approach facilitates more targeted hypotheses adopted in [9, 15, 22, 28].

Other variations alter the model input instead of the **model representation** or hidden states. For instance, Gao et al. [19] leverage attention weights to capture word-to-word relations rather than isolated representations, while Rahimi et al. [16] use attribution-based features from explainable AI methods to quantify each preceding word's impact on next-word prediction. Finally, several studies replace encoding with representational similarity analysis (RSA) [49], which compares pairwise representational distances between the two systems [1, 13, 32].

## 4 The Platonic Representation Hypothesis

Huh et al. [10] introduced the **Platonic Representation Hypothesis**, which assumes the existence of an abstract, ideal representation space that reflects the true statistical structure of the world. Observable data, including images $(X)$, texts $(Y)$, sounds, etc. are regarded as projections or partial observations of this latent reality. The authors hypothesise that as models scale and improve, their internal representations converge toward this space across architectures, tasks, and modalities. To test this, they use a kernel-based alignment metric to measure whether models place similar data points close together in feature space. If two models place similar data points close together in their feature spaces, they are considered aligned. Rather than coincidence, convergence across modalities suggests that models approximate the world's underlying structure.

The authors provide theoretical and empirical evidence of convergence and propose different explanations for its occurrence. Our review centers on model scale, competence, training, and the stronger convergence observed in multimodal models. Building on this, we consider **brain–model alignment**, which the authors note only briefly as additional support for their hypothesis. Unlike Huh et al. [10], who mention this connection in passing, we directly test whether the factors proposed to enhance convergence likewise predict stronger alignment with biological brains. In other words, we reverse the logic: if the hypothesis is correct, and these factors push models toward an *ideal* representation of reality, and if biological brains share such a representation, then we should observe a positive relationship between each factor and brain-model alignment, Figure 2.

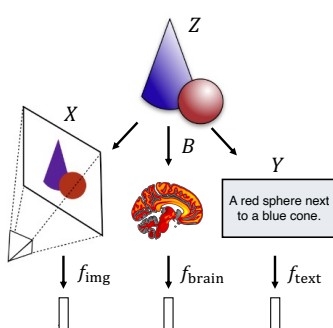

Figure 2: Platonic Representation Hypothesis (adapted from original [10]): Images $(X)$, text $(Y)$, and brain activity $(B)$ are projections of a common underlying reality $(Z)$.

Below, we describe these factors and the evidence found in the works we covered. Concretely, larger models should align better with brain data (Subsection 4.1), models trained on a broader set of tasks should do the same (Subsection 4.2), and multimodal models should align more closely than unimodal ones (Subsection 4.3).

### 4.1 Performance and Scaling Drives Convergence

Huh et al. [10] show that convergence increases with model competence, as higher-performing models show more similar internal structures. Deep networks have an inductive bias toward simple, statistically efficient solutions, and scaling in parameters, data, or compute amplifies this bias, reducing the space of possible representations. Larger models therefore tend to converge on shared representations that mirror the structure of the data. Although scaling alone can induce alignment, different architectures vary in how effectively they exploit it. Overall, growing scale leads models toward increasingly aligned representations that approximate a shared statistical model of the world which leads us to propose Hypothesis 1.

**Hypothesis 1** *Larger and more capable models should align more strongly with brain activity.*

Table 2: Overview of datasets and models employed across reviewed studies. Rows are grouped and color-coded by modality ( speech , text, speech , text , images, text , video, text , and multimodal )

| Reference | Dataset | Model(s) |
|---|---|---|
| [21] | Passive natural language listening [50] (L) | Wav2Vec2.0 [41] and HuBERT [42] |
| [22] | Podcast Stories [51] (L) | Wav2Vec2.0 [41] and HuBERT [42], Whisper [52] |
| [28] | Subset Moth Radio Hour [53] (R) | BERT [35], GPT-2 [36], T5 Flan [54], Wav2Vec2.0 [41], Whisper [52] |
| [7] | Podcast Stories [51] (L) | OPT [55], LLaMA [37], HuBERT [42], WavLM [56], Whisper [52] |
| [26] | The Little Prince [57] (L) | Monolingual, multilingual, untrained BERT [35], Whisper [52] |
| [24] | Harry Potter Dataset [58] (R) | BART [39], LED [59], BigBird [60] and LongT5 [40] |
| [15] | Narratives [61] (L) | BERT [35], GPT2 [36] |
| [25] | Pereira [62] (R), BLANK2014 [63] (L), Harry Potter Dataset [58] (R) | GPT2 [36], T5 [40], LLaMa 2 [37], Vicuna, Alpaca [64], T5 Flan [54] |
| [5] | Harry Potter Dataset [58] (R) | GPT-2 [36] |
| [20] | Pereira [62] (R+V) | GPT-2 [36] |
| [14] | Pereira [62] (R) | GPT-2-XL [36] |
| [23] | Moral judgement [65] | BERT [35], DeBERTa [66](T), RoBERTa [67] |
| [6] | Podcast Stories [51] (L) | OPT[55], Pythia [68] |
| [17] | The Little Prince [57] (L) | Llama 3 [37], Gemma [69], Baichuan2 [70], DeepSeek-R1 [71], GLM [72], Qwen2.5 [73], OPT [55], Mistral [74], BERT [35] |
| [18] | Natural Stories fMRI [75] (L), Pereira [62] (R) | GPT-2 [36], GPT-Neo [76], OPT [55], and Pythia [68] |
| [27] | Moth Radio Hour [77] (R) | Monolingual (text english, chinese), multilingual BERT [35], XLM-R, XGLM, LLaMA-3.2 [38]) |
| [16] | Narratives [61] (L) | GPT-2 [36], LLaMA 2 [37], and Phi-2 [78] |
| [19] | Reading Brain [79] (R) | LLaMA [38], GPT [36], Mistral [74], Alpaca [64], Gemma [69] |
| [29] | Sherlock clips [29] (L+V) | ViT [43], Word2Vec [80], GPT2 [36] |
| [8] | Natural Scenes Dataset [81] (V) | InstructBLIP [45], mPLUG-Owl [46], IDEFICS [82], ViT-H [43], and CLIP [44] |
| [32] | Pereira [62] (R+V) | GPT-2 [36] , Qwen-2.5 [73], Vicuna-1.5 [83], FLAVA [84], LLaVA [85], Qwen2.5-VL [86] |
| [30] | Moth Radio Hour [77] (R), Movie watching [87] (L+V) | BridgeTower [88], RoBERTa [67] and ViT [43] |
| [31] | Japanese movie [31] (L+V) | Word2Vec [80], BERT [35], GPT2 [36], OPT [55], Llama 2 [37], CLIP [44], GIT [89], BridgeTower [88], LLaVA [85] |
| [13] | BOLD Moments Dataset [90] (V) | ResNet-50 [91], ViViTB [92], CodeLlama-7B, Llama3-8B [38], BLIP-L [93], LLaVA-OV-7B [94] |
| [9] | Movie10 [33] (L+V) | ImageBind [95], TVLT [96], Wav2Vec2.0 [41], ViT-B [43], ViViTB [92], VideoMAE [97] |

The review by Karamolegkou et al. [3], focusing on earlier work, reported that brain-model similarity increases modestly with model size. Building on this, we examine recent studies for additional evidence. Antonello et al. [7] investigated how model architecture, size, and training data influence the ability of LLMs to predict human brain activity, showing that the brain prediction performance scales logarithmically with model size. Their results further indicate that increases in training data volume and downstream task performance correlate with improved neural alignment, whereas increasing hidden state size without corresponding performance gains can actually degrade encoding quality. Consistent with this trend, both Lei et al. [17] and Gao et al. [19] found that larger models exhibit better alignment with brain responses.

Challenging this view, Lin et al. [18] tested whether brain alignment reflects true linguistic learning or simply the artifact of larger vector dimensionality. Comparing trained models to untrained but dimension and architecture-matched counterparts, and modelling only residual variance, they found that controlling for dimensionality diminishes or even reverses the apparent scaling benefit: larger models do not align better, and the unique contribution of training may even decrease with size. In contrast, Oota et al. [15], through residual analysis (Subsection 3.3), showed that specific linguistic properties make a genuine contribution to brain alignment.

In parallel, Merlin and Toneva [5] demonstrated that brain alignment cannot be explained by next-word prediction performance alone. Even when controlling for word-level information and prediction accuracy, residual alignment persists in language regions, suggesting that models capture additional properties relevant to brain responses. Complementing these perspectives, Antonello and Cheng [6] provided evidence that alignment increases with training through a two-phase abstraction process, in which intermediate layers construct higher-dimensional, compositional representations. Similarly, Hosseini et al. [20] showed that models trained on developmentally realistic data volumes (100M words) already achieve near-maximal alignment, comparable to models trained on billions of words, with further gains in prediction accuracy failing to enhance alignment. Taken together, these findings support Hypothesis 1: Although alignment tends to increase with model scale and performance, neither scaling nor predictive coding performance fully account for it, suggesting that alignment relies on richer representational mechanisms beyond word-level prediction.

## 4.2 Task Expansion Drives Convergence

Another dimension of convergence highlighted by Huh et al. [10] is task diversity. Training on a broader set of tasks forces models to find representations that satisfy multiple objectives. As task diversity increases, the space of viable representations shrinks, pushing models toward shared, general-purpose representations, since every new task or dataset adds constraints. The paper visualises this as the intersection of constraint regions in representation space. Based on this, we propose Hypothesis 2

**Hypothesis 2** *Models trained on a broader set of tasks should align more strongly with brain activity.*

Aw and Toneva [24] demonstrated that fine-tuning on a narrative summarization task, BookSumf [98], produces richer and more brain-like representations than training models exclusively on generic web data, despite not improving language modeling performance.

Another line of work focused on **instruction fine-tuning**, where models are trained on many hetero-geneous tasks phrased as natural language instructions (e.g., summarization, Question Answering (QA), classification). Such training is thought to push models closer to an idealised Platonic represen-tation, by optimizing under diverse constraints. Aw et al. [25] evaluated 25 models and reported that instruction-tuned models improve alignment by an average of 6%. Consistently, Lei et al. [17] found that instruction-tuned models outperform their base counterparts, whereas Gao et al. [19] showed that, when controlling for size, instruction-tuning yields no significant benefit.

An increasing number of studies have explored **brain-tuning**, i.e., fine-tuning models with fMRI data. Conceptually, this adds an additional constraint, forcing models to align not only with external objectives (e.g., language modelling, image classification) but also with biological representations. Within the framework of the PRH, such constraints further narrow the representational space, pushing models toward a shared, modality-independent structure. For example, Meek et al. [23] tested whether fine-tuning encoder-based LLMs on moral reasoning data or directly on fMRI recordings improves neural alignment, using the ETHICS benchmark [99] and the Moral Judgments dataset [65]. Their results indicate that neither strategy consistently improves brain alignment or task performance, suggesting that targeted fine-tuning alone may be insufficient.

Regarding text and audio models, Oota et al. [28] investigated alignment during reading and listening, emphasizing the contribution of low-level features. Their findings indicate that text-based models achieve stronger alignment overall, particularly in late language regions where alignment reflects semantic rather than stimulus-driven processing. Moreover, text-based models exhibited superior cross-modal transfer (e.g., to visual and auditory regions), implying that they encode richer and more generalisable representations than the speech-based ones. Building on this work, Moussa et al. [22] applied brain-tuning to speech models, improving alignment with semantic regions, reducing reliance on low-level features, and enhancing downstream semantic performance without impairing speech abilities. Moussa and Toneva [21] confirmed and extended these results, showing that brain-tuning

reorganizes representations into a clearer progression from acoustics to semantics. Together, these findings suggest that while text models initially align more closely with brain processing and may approximate the modality-independent space proposed by the PRH, brain-tuning moves speech models closer to this representation.

Finally, Negi et al. [27] showed that multilingual LLMs outperform monolingual ones in brain alignment and cross-lingual transfer, even without fine-tuning. Fine-tuning on brain data yields only small, inconsistent gains, suggesting that the main advantage stems from multilingual pre-training rather than brain-based fine-tuning.

### 4.3 Cross-Modal Training Drives Convergence

The authors show that better LMs tend to align more strongly with vision models like DINOv2 [100], and that multimodally trained models such as CLIP exhibit higher vision–language alignment, which drops when fine-tuned on a single modality. Cross-modal training encourages models to learn representations that are not tied to a single modality but capture a shared, abstract structure—bringing them closer to a *Platonic* representation. Using diverse data types, such as image–text pairs, constrains models to discover representations valid across modalities. This motivates the following Hypothesis 3.

**Hypothesis 3** *Models trained on more modalities should align more strongly with brain activity.*

Most prior studies have compared **VLLMs** with LLMs. Oota et al. [8] showed that instruction-tuned MLLMs align slightly better with brain activity than CLIP, with both outperforming vision-only models, thus supporting the benefit of cross-modal integration. Similarly, Tang et al. [30] found that multimodal transformers generalise across modalities (e.g., trained on movies and applied to speech), with multimodal features showing stronger alignment in higher-level cortical regions. Small et al. [29] further reported that multimodal embeddings outperform unimodal ones in language and social brain regions but not in visual areas, suggesting non-uniform convergence. Along these lines, Nakagi et al. [31] showed that multimodal vision–semantic models better explain high-level narrative regions, with PCA isolating variance linked to semantic features such as background story. Finally, Ryskina et al. [32] confirmed the advantage of VLLMs for cross-modal conceptual meaning and introduced novel conceptual ROIs-voxels that respond consistently to the same concepts across modalities, best predicted by both LLMs and MLLMs. This supports the view that models capture modality-independent conceptual information, approximating a *real world* representation.

Zada et al. [26] provided evidence from languages and **audio**, showing with fMRI from monolingual speakers that unilingual BERT embeddings are similar but rotated, aligning more closely for related languages and predicting comprehension activity across languages with minimal loss. In contrast, multilingual and multimodal models captured more abstract, language-independent concepts: their mid-layers are less language-specific, and alignment is stronger for languages closer to the native tongue. These results suggest that broader linguistic and modality exposure yields richer conceptual spaces that better align with the brain. Extending beyond language and audio, Han et al. [13] introduced **video** as a modality and reported that image–language and video–LMs show stronger alignment in higher-level brain regions, with predictivity rising from early to later layers, especially when models integrate predictive processing across modalities. Similarly, Oota et al. [9] showed that video–audio models outperform unimodal ones across language, visual, and auditory regions. Importantly, this effect extends beyond low-level sensory features: cross-modal training enhances alignment in language areas, reinforcing the view that multimodal exposure yields more brain-like, world-grounded representations. Overall, these findings support Hypothesis 3, indicating that cross-modal training pushes models toward modality-independent conceptual representations.

## 5 The Intermediate Layer Advantage Hypothesis

Skean et al. [11] build on prior work showing that linguistic and semantic features often emerge in middle layers, while final layers become overly tuned to pretraining in LMs. On the one hand, they introduce a unified framework—combining information-theoretic, geometric, and invariance-based metrics (e.g., DiME [101], curvature [102], InfoNCE [103])—to evaluate layer-wise representation quality across architectures, model sizes, and tasks. On the other hand, their empirical results show that intermediate layers consistently outperform final ones on 32 MTEB benchmark tasks [104], sometimes by up to 16%, a trend observed in both Transformers (Pythia [68], Llama3 [38], BERT [35])

and State Space Models (SSMs) (Mamba [105]). Importantly, their metrics not only correlate strongly with downstream performance but also peak at the same intermediate layers, thereby providing both an empirical and theoretical demonstration that these layers yield the most robust and generalizable representations. They also identify architecture-specific patterns: autoregressive decoders show a pronounced mid-layer compression valley, while bidirectional encoders remain more uniform. Extending to vision, only autoregressive image transformers, like AIM [106], display the same mid-depth bottleneck, suggesting the training objective, not the modality, drives this effect.

**Hypothesis 4** *If intermediate layers of LMs encode the most robust and generalizable linguistic and semantic features, then these layers should also show the strongest alignment with brain activity.*

Toneva and Wehbe [12] first demonstrated that middle layers of LMs show the strongest alignment with brain language regions, a finding repeatedly replicated in subsequent work. This observation aligns with recent evidence that middle layers encode richer and more generalizable linguistic representations [11]. Here, we assess whether the studies under review provide additional support for this pattern of layer-wise convergence across model classes.

Several studies focused on **decoder**-based **text** LMs. Of particular relevance, Antonello and Cheng [6] examined a question directly related to the hypothesis in Skean et al. [11]. They found evidence that LMs undergo a two-phase abstraction process during training, an early composition phase, and a later prediction phase, reflected in how well the model layers align with brain activity. Results show that layers with higher intrinsic dimensionality exhibit stronger brain alignment and that such brain-aligned representations emerge predominantly in the middle layers. Moreover, as training advances, the composition phase becomes compressed into fewer layers, suggesting that training simultaneously improves task performance and sharpens the emergence of cognitively relevant representations.

Similar evidence is described in Antonello et al. [7], comparing LLaMA and OPT. The LLaMA models are marginally better at encoding than the OPT models and reach peak performance in relatively early layers followed by a slow decay. In contrast, OPT models achieve their maximum performance in layers that are roughly three-quarters into the model, which mirrors results observed in other decoder models. They propose the larger training set of the LLaMA models as an explanation for both their superior encoding performance and their different layer-wise pattern. Other studies (e.g., Lei et al. [17], Kauf et al. [14]) find that many models also peak in intermediate layers.

Complementing this, Rahimi et al. [16] reached a similar conclusion, focusing on importance rather than representations (i.e., how much each word in context contributes to next-word prediction). Early layers correspond to initial stages of language processing in the brain, while later layers align with more advanced stages. Layers with higher attribution scores, i.e., more influential for prediction, also show stronger alignment with neural activity.

In **encoder–decoder** text models, Aw and Toneva [24] evaluated four architectures and found that improvements emerge in intermediate layers or are distributed across depth before and after fine-tuning, but never peak in the final layer. Similarly, Oota et al. [28] showed that text-based models follow a clearer progression from low- to high-level representations: early layers encode superficial textual features (e.g., number of letters or words), which diminish in deeper layers, where alignment with later-stage brain regions strengthens. For these models, alignment peaks in mid-to-late layers, consistent with the emergence of abstract, semantically relevant representations.

Additionally, several studies investigated **encoder**-based text model architectures. Specifically, Oota et al. [15] found that in all cases performance is strongest in the middle layers. Similarly, Zada et al. [26] analysed monolingual and multilingual models—for monolingual models, correlations between unimodal embeddings increased through the late-intermediate layers and dropped in the final layer, while for multilingual models, cross-language correlations grew until roughly three-quarters depth before declining. These findings suggest that the first and last layers are associated with language-specific processes, whereas intermediate layers capture more conceptual representations.

For **audio** models, Oota et al. [28] analysed layer-feature correlations and found that speech-based models follow a different pattern from text-based ones. Low-level features such as Mel spectrograms, and phonological information are strongly encoded in the early and intermediate layers and persist even in deeper layers. These features drive alignment with early auditory cortices, but do not support robust alignment with late language regions once removed, indicating that speech models retain a sensory-phonological focus and lack abstract semantic representations in deeper layers. Building on this, Moussa and Toneva [21] showed that brain-tuning reshapes this hierarchy: early layers remain

acoustic, while late layers align strongly with higher language regions and capture complex semantic information, closely mirroring the brain's progression from acoustics to semantics.

In a different vein, Antonello et al. [7] reported that upper-middle and uppermost layers generally yield the best performance, except in Whisper, where performance increases steadily with depth, likely due to the use of only the encoder. Complementing this, Zada et al. [26] showed that when both Whisper modules are used, correlations between embeddings of different languages peak in the encoder's final layers and in the decoder's mid-to-late layers.

Comparing text and **vision** models, Han et al. [13] observed that text models show weak alignment in early layers but stronger alignment in middle and final layers, mainly in language and higher cognitive regions. In contrast, vision models align strongly with early and mid-visual areas but alignment diminishes at later stages. Multimodal models demonstrate broad alignment from the outset, sustain it across middle layers, and in final layers LLaVA (a predictive multimodal model) achieves strongest overall alignment, while BLIP-L shows reduced alignment in vision. Training objective plays a significant role: classification and captioning models align better at early layers but weaken at depth, whereas predictive models improve in middle and late layers, maintaining or enhancing alignment in higher cognitive regions, especially under multimodality. Building on these findings, Tang et al. [30] reported that BridgeTower achieves peak accuracy in intermediate layers associated with cortical conceptual regions, while Oota et al. [8] showed that InstructBLIP and IDEFICS align with higher visual regions in middle layers and early visual regions in later layers, whereas mPLUG-Owl reaches maximal alignment in late layers across both high- and low-level regions.

Finally, Oota et al. [9] extended this comparison to **video** models, reporting consistent declines in performance from early to deeper layers, for both multimodal and unimodal models. Their key finding is that joint video-audio embeddings achieve superior brain alignment across all layers relative to unimodal video or speech embeddings.

# 6    Conclusions

This field has gained momentum, as reflected in the surge of studies over the past two years. Multiple factors have been examined as potential drivers of brain-language model alignment, with growing attention to multimodal models, particularly VLLMs. According to the PRH, such alignment arises because improving models approximate the same underlying representation of the world that the brain interprets. The studies we reviewed generally support this view: alignment tends to be stronger for larger, higher-performing models and those trained on more tasks (Subsection 4.1). Fine-tuning likewise improves alignment, consistent with the claim that adding data, tasks, or constraints drives representational convergence. Evidence that brain-tuning or human alignment brings models closer to this representation, however, appears weaker (Subsection 4.2). Cross-modal training, by contrast, reliably enhances alignment by pushing models toward modality-independent conceptual representations. Overall, higher-performing models align more broadly across the brain, including in regions not tied to their training modality. By testing evidence from the original hypothesis against an additional line suggested by Huh et al. [10], brain–model alignment, we find converging support for the existence of such an underlying representation, though the evidence is not uniformly positive.

At the same time, evidence shows that final layers are not those that align most strongly with the brain (Section 5), in line with Hypothesis 4. Across architectures and brain regions, studies vary, but a general trend emerges: intermediate layers yield better alignment with relevant regions. This too resonates with the PRH: intermediate layers appear to encode representations closer to the world's underlying structure, while final layers are increasingly specialised to pre-training objectives and less universally aligned. Finally, Table 3 provides a summary of the qualitative analysis of how the reviewed works support (or contradict) the hypotheses proposed in this review. We considered a study to show agreement or disagreement when its findings could be qualitatively related to a given hypothesis. Cases labelled as neutral correspond to studies whose results could not be clearly interpreted as either supporting or contradicting the hypothesis. Framing alignment through the dual lenses of the PRH and the Intermediate-Layer Advantage provides a theoretical scaffold for designing future experiments, benchmarks, and models that more directly probe shared human–artificial representational structure.

**Limitations.** The studies we analysed differ in metrics, datasets, and alignment protocols, as also noted by Karamolegkou et al. [3]. Such heterogeneity implies that effects of scale or layer depth

Table 3: Overview of the reviewed studies classified by modality ( speech , text, speech , text , images, text , video, text , multimodal ). This classification follows the same organisational structure as Table 2 to facilitate direct comparison. Each column represents one of the four hypotheses introduced in Section 4 and Section 5. Cell colours convey the qualitative degree of support: strong disagreement , disagreement , neutral , agreement , and strong agreement .

| Reference | Hypothesis 1 | Hypothesis 2 | Hypothesis 3 | Hypothesis 4 |
|---|---|---|---|---|
| [21] |  |  |  | strong agreement |
| [22] |  | agreement |  |  |
| [28] |  |  |  |  |
| [7] | strong agreement |  |  |  |
| [26] |  | agreement |  |  |
| [24] |  | agreement |  | strong agreement |
| [15] | agreement |  |  | strong agreement |
| [25] |  | agreement |  |  |
| [5] |  |  |  |  |
| [20] | agreement |  |  |  |
| [14] |  |  |  | strong agreement |
| [23] |  | disagreement |  |  |
| [6] | agreement |  |  | strong agreement |
| [17] |  | agreement |  |  |
| [18] | strong disagreement |  |  |  |
| [27] | agreement | agreement |  |  |
| [16] |  |  |  | agreement |
| [19] | agreement | disagreement |  |  |
| [29] |  |  | strong agreement | agreement |
| [8] |  |  |  |  |
| [32] |  |  | strong agreement |  |
| [30] |  |  | strong agreement | agreement |
| [31] |  |  | agreement |  |
| [13] |  |  | agreement |  |
| [9] |  |  | strong agreement |  |

should be treated as qualitative patterns rather than quantitative laws. More broadly, interpretation is complicated by choices of stimuli, resolution, recorded brain regions, and neural preprocessing, all of which capture different aspects of cognition. Architectural details (e.g., tokenization in language, visual preprocessing in vision) may further interact with the data. Our review emphasises general trends across 25 language-related studies without attempting to parse finer-grained regional or processing differences. Vision-specific work, such as Gifford et al. [107], thus lies outside our scope.

## Acknowledgments and Disclosure of Funding

This research is supported by Horizon Europe's European Innovation Council through the Pathfinder program (SYMBIOTIK grant 101071147) and by the Industrial Doctorate Plan of the Department of Research and Universities of the Generalitat de Catalunya Grant AGAUR 2023 DI060.

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

# A  Appendix

## A.1  Complementary details of the reviewed works

Table 4: Models grouped by modality.

| Modality | Models |
|---|---|
| Text | BART [39], LED [59], BigBird [60], LongT5 [40], Word2Vec [80], GPT-2 [36], OPT [55], LLaMA 2 [37], GPT-Neo [76], T5 [40], Vicuna [83], Alpaca [64], T5 Flan [54], GPT-2-XL [36], DeBERTa [66], RoBERTa [67], Pythia [68], LLaMA 3 [37], Gemma [69], Baichuan2 [70], DeepSeek-R1 [71], GLM [72], Qwen2.5 [73], Mistral [74], BERT [35] (monolingual and muntilingual), XLM-R, XGLM, LLaMA-3.2 [38]), Phi-2 [78] |
| Speech | Wav2Vec2.0 [41], HuBERT [42], WavLM [56] |
| Image | ViT-H [43], ViT [43], ResNet-50 [91] |
| Video | VideoMAE [97], ViViTB [92] |
| Image + Text | InstructBLIP [45], mPLUG-Owl [46], IDEFICS [82], CLIP [44], BLIP-L [93], BridgeTower [88], GIT [89], LLaVA [85], FLAVA [84], Qwen2.5-VL [86]. |
| Video + Text | LLaVA-OV-7B [94] |
| Video + Speech | TVLT [96] |
| Speech to Text | Whisper [52] |
| Multimodal | ImageBind [95] |

