# OpenReview forum: "Brain–Language Model Alignment: Insights into the Platonic Hypothesis and Intermediate-Layer Advantage"
_NeurIPS.cc/2025/Workshop/UniReps — UniReps2025_

### Official Review · Reviewer_8PA6 · 2025-09-15
**Review of survey, some limitations but fulfills a good niche in the current literature**

**Confidence:** 4

**Review:**

**Summary**:

This paper presents a survey of recent work in the field of brain-model alignment, taking care to frame the reviewed papers in terms of two hypotheses concerning internal representations, the Platonic Representation Hypothesis (PRH) and the Intermediate Layer Advantage Hypothesis (ILAH) respectively. The survey consists of 25 different recent works, and there is a qualitative analysis of the results in terms of differing data modalities, training objectives, scale, etc., in line with the general movement of the field concerning what different aspects of training and data might lead to different types of representations. Of particular interest, the authors find that many different axes along which models might be measured contribute to increased brain likeness, and that the intermediate layers of these models tend to be the most brain like. These findings support both of the hypotheses in question.

While there have been other surveys on brain likeness, this short survey presents a different framing and takes care to analyze recent advances in fMRI studies only. Likewise, this study does not rise to the level of a true meta-analysis, due to lack of statistical examination of results between the different studies. However, this is difficult to do due to the many confounding factors between each study, such as lack of similar datasets and comparison methods. This is to be expected, as the field of brain-model alignment is still not truly mature and there is a lot of ongoing work to find the best methods for calculating brain likeness, which neuroimaging modalities are best for comparison, etc. So the study is constrained to be qualitative in nature. There were some things lacking (covered in detail in the limitations section below), such as a more opinionated review of research finding null results for these two hypotheses, but ultimately this survey occupies a nice niche in the literature and combines together related hypotheses that haven't yet been surveyed in this way (to the best of my knowledge). Therefore, I recommend acceptance.

**Strengths and Weaknesses**:

1. Quality
	1. 3/4. I think that there are some minor things that make this an imperfect paper, which I list in the limitations section. For what it's worth, I believe that these are all fixable and the paper still deserves acceptance.
2. Clarity
	1. 4/4. The authors do very well in their framing and analysis of the literature.
3. Significance
	1. 3/4. In my opinion, this paper occupies a nice niche in the field, but ultimately is still small in nature. This is not a major concern seeing how this is still a rapidly growing area, and 25 different fMRI on this is still a fairly good sampling.
4. Originality
	1. 3/4. This paper is connecting different lines of thought together, but the inferential leap is not massive. However the paper still deserves praise for bringing these lines of thought together in a methodical fashion.

**Questions & Limitations**:
Note that I am combining the questions and limitations section of the NeurIPS reviewer guideline into one, because there is no rebuttal period for the workshop.

So there are a few different concerns I have with the paper, both major and minor. Keep in mind that I am recommending acceptance, but these could still be fixed.

Minor concerns:
- In the introduction, you introduce 'Language Models (LMs)' and 'Large Language Models (LLM)' where both are in bold. These are in lines 16 and 37. I believe that the authors mean the same thing here, so just make sure that this is not redundant.
- Some of the background highlighting color choices in the different tables are very difficult for me to make out. I turned on 'dark mode for content' in my PDF reader and only then saw that everything had some type of background in the leftmost column of Table 2, for instance.
- There is a typo in the caption of Figure 2: 'centres' -> 'centers', when it's a verb. Other parts of the caption also seem redundant in phrasing: 'the authors note only briefly', immediately followed by '... who mention this connection in passing ...'.
- On line 136, turn 'propose the Hypothesis 1' into 'propose Hypothesis 1'.

Major concerns:
- In the third paragraph of section 4.1, and other areas, you briefly summarize many papers who all give contradicting or null results. It's fine to do a more *qualitative* survey, but in that case you should then provide some editorial opinion on which is correct or what potential confounding factors could be causing this among the different papers you've cited. This would help provide more guidance for downstream research. As the authors of the survey, you're the most well positioned to give takes on these phenomena such that the community can come to a better world model of the problem. It's fine to qualify this with some uncertainty too.
- Similarly, you make the case in this and other areas, that the general tendency is in support of the two hypotheses you frame your survey around. For instance, saying that scaling is not sufficient by itself is good, but perhaps it is best to frame things in terms of necessary and sufficient conditions for better clarity. When speaking in generalizations, a more formal framing like that might not be possible, but when you introduce a tendency then you can still make the uncertainty clear in the stated hypothesis.
- Some of the paper introductions follow a pattern where each paper is given a one sentence summary, and the bulk of some paragraphs is given over to this style of literature review. It seems more valuable to approach this from a higher level, systematic perspective where you discuss several related papers and the broad takeaways of them, only calling out each individually when there is something profoundly different.
- You introduce four main hypotheses, discussing the PRH and ILAH, and the different tendencies of things like scale. I understand that the workshop constrains paper length by a lot, but I think it would be very conceptually valuable to have a table listing out all the different studies you've reviewed (similarly to table 2), and then you can have each of these four hypotheses as a column and mark the level of support each study gives to each hypothesis. That would be a much more succinct way of presenting this information, and would help the reader to immediately get at your point.

I think that overall these limitations are not deal breakers, and the strengths of your paper are still enough for me to recommend acceptance. Good work! But in the future, a bit more care paid to these details would help you make your point to the reader better.

**Overall Score**:

4/5, Accept.

**Confidence**:

4/5.

**Score:**

4

**Topic Fit:**

3

---

### Official Review · Reviewer_x6vA · 2025-09-16
**Timely and Well-Organized Review Paper Synthesizing fMRI–LM Alignment Evidence in Support of Two Widely Discussed Hypotheses**

**Confidence:** 3

**Review:**

**Summary** -
The paper provides a comprehensive review of recent fMRI-based studies (2023–2025) that investigate alignment between language models and human brain activity. The authors focus on two key organizing frameworks: the Platonic Representation Hypothesis (PRH) and the Intermediate Layer Advantage Hypothesis. While the primary modality examined is language, the review also discusses emerging multimodal and cross-modal evidence—particularly in support of the multimodality/ amodal convergence argument within PRH. Through cross-study comparison and synthesis, the authors conclude that these studies generally support both frameworks.


**Strengths**

- Clarity & Organization: The paper is well-structured and well-accessible to readers with familiarity and interest in LM-brain representation alignment. The two main hypotheses are clearly presented, and the summary tables were effective in abstracting shared properties (questions, shared methodology, data types etc) from different POVs among the 25 studies.

- Comprehensiveness: The paper makes a strong attempt to aggregate and contrast evidence from diverse model families and across a wide range of evaluation methods. It is also timely in addressing the role of PRH in model–brain alignment, an area of active interest esp to UniReps audiences. This synthesis provides a useful “hub” for recent literature and enables cross-study comparison framed through the lenses of PRH and ILA.

- While not fully developed, the paper commendably attempts to integrate ILA into the broader framework of PRH (as in the framing of Hypothesis 4 and lines 350–352 in the conclusion), albeit the connection could be made more straightforward.

**Weakness** -
- Limited Interrogation of Assumptions & Potential Competing Hypotheses (if any): The review presents the two central hypotheses as relatively cohesive but does attempt to mention their potential falsifiability or potential competing hypotheses that could potentially explain the evidence reviewed (e.g., What empirical results might challenge their validity? Could the same evidence be explained by competing hypotheses, or is PRH the only available explanation?). Admittedly, this may be somewhat demanding for a review paper if the field lacks significant critiques or alternative hypotheses, but if such perspectives exist, engaging with them would strengthen the paper by adding a more critical stance.


- Methodological Heterogeneity: As acknowledged by the authors, the reviewed studies vary significantly across model types, alignment metrics, fMRI preprocessing, and task paradigms. While this diversity offers breadth, it limits the generalizability of conclusions drawn at the level of individual studies. Some exceptions are explained anecdotally rather than being tied to generalizable methodological patterns.

- Limited Analysis of Architecture–Layer–Brain Interactions: While intermediate layers are discussed at length, there is limited synthesis of how alignment patterns vary with architecture type, model scale, or training modality beyond study-specific commentary. These factors likely shape both alignment depth and cortical mapping. For instance, do studies that share one factor but differ in another still show consistent trends, and are such patterns robust when one variable is swapped out? An additional summary table dedicated to the ILA hypothesis—organized along architecture × training modality × alignment layer × brain region, and ideally accompanied by high-level commentary on possible mechanisms—categorizing observed alignment peaks by model type and their best-matching brain regions would be valuable for readers to better draw theoretical insight.


Overall, while some expansions could strengthen the paper (as noted in the weaknesses, though a few may be somewhat demanding), the clear and well-structured synthesis of evidence around two widely discussed hypotheses in brain–LM (or model) representational alignment—one addressing the potential explanation (“why”), another focusing on localization (“where”), and both linked to the role of intermediate layers in providing more generic features (“what”)—fits well with the theme of the UniReps workshop and will be of broad interest. As a review paper, it serves as a useful resource for researchers to follow and cross-compare evidence in this area. I therefore recommend acceptance.

**Score:**

4

**Topic Fit:**

3

---

### Official Review · Reviewer_QLN5 · 2025-09-16
**Inappropriate paper for archival submission**

**Confidence:** 4

**Review:**

This paper aims to present a systematic review of literature that investigate similarity between brain representations and those of Language Models. The authors then move to relate the findings of these papers to the Platonic Representation Hypothesis as well as the Intermediate Layer Advantage. Unfortunately, this paper does not meet the criteria for a rigorous systematic review.

Typically, a systematic review should follow something like the PRISMA 2020 guidelines, available here https://www.prisma-statement.org/ . OSU has a great, high-level explanation here: https://hslguides.osu.edu/systematic_reviews/steps .

In short, the authors do not clearly delineate their search and inclusion criteria for reviewed papers, do not present a standardized data extraction protocol from included papers, and do not clearly state a statistically verifiable hypothesis regarding how evidence may or may not support the PRH or ILA.

In general, this work would be more appropriate for the non-archival track, as it is more of an opinion piece than systematic review or meta-analysis. As it stands, this submission is the beginning of a strong scoping review (where ) and not appropriate for archival submission.

**Score:**

1

**Topic Fit:**

2

---

### Official Review · Reviewer_xt1W · 2025-09-17
**Review 30**

**Confidence:** 4

**Review:**

**Abstract**: The abstract provides a clear overview of the paper’s aims: reviewing 25 recent fMRI-based studies to test two theoretical hypotheses (the Platonic Representation Hypothesis and the Intermediate-Layer Advantage). It is concise and focused, though it could be sharpened by explicitly highlighting the *novel contributions* of this review relative to existing surveys (e.g., why focusing on 2023-2025 fMRI work changes the conclusions).

---

### Strengths
- **Timely and relevant**: Covers a surge of recent (2023-2025) brain-LM alignment studies, ensuring up-to-date synthesis.
- **Clear theoretical framing**: Anchors the review in two hypotheses (PRH and Intermediate-Layer Advantage), which provides conceptual coherence.
- **Structured synthesis**: Organizes reviewed works by themes (scaling, task-specific training, multimodality, etc.) and presents detailed methodological comparisons (encoding models, RSA, residuals).
- **Integration across modalities**: Highlights text, speech, vision, and multimodal models, enriching the perspective.
- **Balanced interpretation**: Discusses supporting and contradictory findings, noting that evidence is not uniformly positive.

---

### Weaknesses
- **Limited originality**: The paper is primarily a review and does not introduce new experiments, datasets, or quantitative meta-analysis; thus its contribution is mostly synthetic.
- **Evidence evaluation**: While thematic synthesis is strong, the paper does not provide quantitative comparisons (e.g., effect sizes, meta-analytic metrics) across the 25 studies.
- **Methodological heterogeneity**: Although acknowledged, the review could do more to standardize comparisons (e.g., by presenting a unified table of alignment scores across studies).
- **Platonic framing**: The invocation of the Platonic Representation Hypothesis risks philosophical overreach unless tethered more tightly to operational definitions.
- **Layer-wise claims**: The section on the Intermediate-Layer Advantage could be strengthened with more quantitative summaries (e.g., distribution of peak alignment layers across studies).

---

### Detailed Comments

1. **Clarity and Structure**
   - The manuscript is well organized.
   - However, some sections (Conclusions) repeat claims without adding novel insight. Consider streamlining.
   - Define technical terms (e.g., “alignment,” “Platonic representation”) operationally early in the paper to avoid ambiguity.

2. **Contribution and Originality**
   - The main contribution is reframing recent fMRI-LM alignment findings under two theoretical lenses.
   - This is useful, but could be deepened by including **new analyses** (meta-analysis, effect size aggregation, or benchmark proposals).
   - Novelty is therefore limited but meaningful as a focused, theory-driven survey.

3. **Methodology**
   - Strong in terms of cataloging methods (encoding models, residual analysis, RSA).
   - Missing: a systematic coding scheme for reviewed studies (e.g., inclusion criteria, sample sizes, region-of-interest focus).
   - Suggest adding a supplementary table coding each study by: dataset, model, method, sample size, ROI, alignment metric, key finding.

4. **Data and Analysis**
   - The review remains qualitative; no quantitative synthesis of effect sizes or confidence intervals.
   - A forest-plot style meta-analysis would greatly strengthen the contribution, even if limited to correlations reported.

5. **Results and Discussion**
   - The paper makes strong claims: that evidence generally supports both PRH and Intermediate-Layer Advantage.
   - However, the evidence is mixed (e.g., scaling does not always improve alignment, brain-tuning shows weak benefits). These nuances should be emphasized more clearly.
   - Expand on *limitations*: differences in preprocessing, fMRI resolution, or ROI selection may confound cross-study comparisons.

6. **References**
   - Generally comprehensive and up to date, but some references are inconsistent in style.
   - Missing key works: e.g., Schrimpf et al. 2021 (integrative modeling), Fedorenko & Blank (language networks), Lake et al. 2017 (cognitive science perspective).
   - Ensure all references (e.g., Vaswani et al. 2017 for Transformers) are correctly cited.

7. **Ethics and Compliance**
   - No major ethical concerns, as the review synthesizes published studies.
   - A short ethics statement (e.g., all data reviewed were from previously approved studies) could be added for completeness.

---
- **On topic & template**: Appears relevant, though formatting/length may exceed page limits. Needs trimming to 4/8 pages excluding references.
- **Novelty**: **Yes (limited)**. Provides new theoretical framing, but not empirical.
- **Clarity of claims**: **Yes (partially)**. Claims are clear but sometimes overgeneralized.
- **Community interest**: **Yes**. High interest for the community given rapid growth in brain-LM alignment research.

**Overall Score**: 3 = Accept (weak accept)
The paper lacks empirical novelty but provides a useful, well-structured synthesis. With added quantitative analysis, improved clarity, and formatting compliance, it could be a good contribution.

---

**Overall Recommendation**: *Revise and accept for workshop.*
The review is valuable and timely, but it would benefit from:
- Stronger quantitative synthesis of reviewed studies,
- Clearer operationalization of “Platonic representation”
- A standardized comparative table, and
- Formatting/conciseness edits.
These changes would elevate it from a narrative survey to a more rigorous, theory-testing review.

**Score:**

3

**Topic Fit:**

2

---

### Official Review · Reviewer_NDjj · 2025-09-17
**Right to the point and informative**

**Confidence:** 3

**Review:**

I enjoyed reading this manuscript. It is timely and provides a clear and well-structured overview of the most recent advances in fMRI–LM-based alignment. Given the rapid growth of this field, publishing reviews at relatively short intervals, as this paper does compared to earlier ones, can be valuable in guiding the research community.

**Score:**

4

**Topic Fit:**

3

---

### Official Review · Reviewer_HoYH · 2025-09-17
**Survey on fMRI to LMs alignment**

**Confidence:** 4

**Review:**

**EVALUATION**

The paper is a survey of the recent flourishing literature on aligning AI models to fMRI data. It provides a summary of the modern literature through the lens of two major thought-shifting perspectives that have been recently proposed: Platonic Representation Hypothesis and Intermediate-Layer Advantage. While sometimes the evidence is contrasting, the authors find (maybe yet non-conclusive) patterns of how and when these hypotheses may hold.
I wish the survey went more in depth on which design choices lead to the improvements reported in each study and also explore more in-depth the conter-arguments scholars raise on these techniques [see for example Lampinen et al. 2025].
I don't find this survey exceptionally revealing. While it could be relevant for the workshop community as a source and gathering point of papers, I do not see much more value in this specific summary. Specifically due to the lack of depth in their analysis of what could elements could promote these hypotheses or go against them. I therefore suggest a weak accept.

Minor weaknesses follow.

**CONCEPTUAL**

Given the platonic representation hypothesis, it is not clear whether better and better performing models should align more and more with brains. Is your underlying hypothesis is that the brain is a highly performing model (or at least better performing that all the LLMs tested)?
How do LLMs that have internet wide knowledge of virtually any topic in 20+ languages perform wrt brains? I assume they are better at many tasks than the average brain. Why wouldn't they start to mis-align as they surpass human performance? (on some specific tasks, information retrieval or translation, I am not implying AGI).

**WRITING**

```L15-16``` "*while AI theory offers new explanations for why such convergence might occur*". This claim either lacks references or is wrong. Indeed I am not aware of any *theory* that explains why, what is currently out in the field is an hypothesis (see Platonic Representation Hypothesis (PRH), which is not theory).
In general, I would be wary of considering the PRH AI-theory.

```L19-22``` The very first results on alignment between deep models and brain data came from invasive neuron recordings in non-human primates [Yamins & DiCarlo 2014, and subsequent papers]. I think to say "*most often via functional magnetic resonance imaging (fMRI)*" ignores a great part of the field. I understand the authors are part of the fMRI field, but the paper title, abstract and introduction seem to be written for a broader audience, hence broader recognition is needed.

```L130-132``` "*Deep networks have an inductive bias toward simple, statistically efficient solutions, and scaling in parameters, data, or compute amplifies this bias, reducing the space of possible representations*". It is unclear where this conclusion comes from.

```L153``` Elucidate further what you mean by "*genuine contribution*"

**Score:**

3

**Topic Fit:**

2